# The Impacts of EU Cohesion Policy on Sustainable Tourism: The Case of POSEUR in Algarve

**Bernardo Valente [1,*] and Eduardo Medeiros [2,*]**

1 Centro de Investigação e Intervenção Social (CIS-IUL), Instituto Universitário de Lisboa (ISCTE-IUL), Avenida das Forças Armadas, 1649-026 Lisboa, Portugal
2 DINÂMIA'CET, 1649-026 Lisboa, Portugal
* Correspondence: valente.berna@gmail.com (B.V.); eduardo.medeiros@iscte-iul.pt (E.M.)

**Abstract:** Sustainable tourism is a main priority of European Union policies, with the aim of mitigating the potential harmful consequences of this sector on a given territory. The main research goal of this article is to better understand the impact of the public investments financed via EU Cohesion Policy in the tourism sector in the Portuguese Algarve NUT2. This paper focuses on the Portuguese Operational Programme for Sustainability and Efficient Use of Resources (POSEUR 20142020), which was an operational programme of a Portuguese framework to implement EU Cohesion Policy funds in Portugal between 2014 and 2020, and was specifically focused on supporting sustainable development processes. The sustainability performance of POSEUR in the Algarve was evaluated in five different dimensions of analysis: (1) low-emissions economy, (2) adaptation to climate change, (3) risk prevention and management, (4) environmental protection, and (5) resource efficiency. The results showed that POSEUR had a low impact in almost all dimensions in the Algarve, apart from the adaptation to climate change dimension, which exhibits a high impact score, mostly due to the support given for carrying out the Plans for Adaptation to Climate Change (PAAC).

**Keywords:** sustainable tourism; low-emissions economy; adaptation to climate change; risk prevention and management; EU Cohesion Policy; POSEUR; Algarve





## 1. Introduction

Climate change and a growing globalization process have created a need for the tourism industry to adapt to new challenges to maintain the quality of the environment to attract tourists [1]. As Hamaguchi [2] points out, air quality is fundamental in preserving or increasing tourist flows, while higher levels of pollutants are directly connected to a reduction in profitability in the tourism industry. As such, it is fundamental for this industry to pursue strategies that aim at improving environmental sustainability in various activities. The concept of sustainability in this study was defined by the Brundtland Report [3], in which it was described as an activity that "meets the needs of tourists and host regions, while protecting and enhancing opportunities for the future" [3]. This definition incorporates the mitigation of several factors to which the tourism industry can be prejudicial, such as the over-exploitation of resources or the cultural problems and social disruption caused by tourist practices [4]. To establish an effective public policy that promotes the sustainable development of activities, there is a need for cooperative government action that engages the international, national, and local stakeholders in the decision-making process [5].

In this context, this article analyses the main impacts of environmental-sustainability-related investments in the Portuguese NUTS II region of the Algarve, over the period 2014–2020. These investments were financed via POSEUR—the Operational Programme for Sustainability and Efficient Use of Resources (Programa Operacional Sustentabilidade e Eficiência no Uso de Recursos 2014–2020)—as part of the EU Cohesion Policy framework

that aims at promoting environmental sustainability by intervening in three interdisciplinary policy areas: (1) supporting the transition to a low-carbon economy, (2) promoting climate change adaptation, risk prevention, and management; and (3) protecting the environment and promoting the efficient use of resources [6]. This programme has direct implications for the NUTS II region of the Algarve and for the tourism performance of the region. Therefore, this study analyses the impact of POSEUR on the tourism activity in the Algarve in five different dimensions: (a) low-emissions economy, (b) adaptation to climate change, (c) risk prevention and management, (d) environmental protection, and (e) resource efficiency. These dimensions were chosen based on POSEUR's three axes of intervention and the overall policy goals of the EU Cohesion Policy framework.

Ultimately, this paper proposes assessing the main impacts of POSEUR (2014–2020) toward the support of sustainable tourism in the Algarve by applying a territorial impact assessment (TIA) methodology (named TARGET_TIA). Ultimately, the analysis proposes answering the following main research questions: (i) "To what extent have POSEUR investments promoted sustainable tourism in Algarve?"; and (ii) "In which of the five analysis dimensions has POSEUR had the most positive contributions regarding sustainable tourism in the Algarve?".

Portugal has established various plans to resist the impacts of climate change, including the PNPOT [7,8], which established guidelines to trace the problems of the Portuguese territory and define new strategies to improve sustainability in the tourism domain. At a regional level, the different PROTs (Regional Spatial Planning Plan—Plano Regional de Ordenamento do Território) contributed to the development of the Portuguese regions, with special concern for the structural vulnerabilities and sensibility of every region.

The Portuguese Tourism Strategy 2027 [9] guidelines established five different pillars of intervention: to (a) value the territories and the communities, (b) boost the economy, (c) foment knowledge, (d) generate networks and connectivity, and (e) highlight the potential of Portugal as a tourist destination. These goals are inseparable from the environmental concerns that mandate tourism enterprises follow the sustainability principles required by international standards on environmental sustainability [10]. More recently, with the tourism sector under threat due to the pandemic outbreak, there was a need to guarantee the active participation of local stakeholders and authorities in this "comprehensive management of sustainability" [11].

NUTS II Algarve, located in southern Portugal, has felt the effects of a mass touristic inflow over the past few decades, mostly of European visitors from Germany and the UK [12,13] who are attracted by the natural wonders (climate, landscapes, beaches, etc.) of the region. This represents an opportunity for the economic growth of the region. However, these massive touristic flows can be a source of disruption to the region due to several potential negative environmental impacts. Some have been identified in the Algarve region over the last few decades. Some examples are increasing coastal erosion [1], water shortages [14], the risk of flooding [15], and the risk of forest fires [16], which can also be applied to the whole country.

In this context, it should be noted that the Algarve region has seen some improvements in the creation of networks to promote and evaluate environmental sustainability processes over recent years [17]. Some of the most meaningful initiatives were: (i) the organization of the Tourism Biennial conferences (2019) by the Turismo de Portugal association; (ii) the formation of a sustainable tourism observatory, which is specific to the Algarve region; and (iii) the Green Key programme, which has been rewarding the most sustainable touristic enterprises in the Algarve. The main goal is to promote an efficient local economy by evolving the use of natural resources and efficiently managing the territory to contribute to the development of tourism in the region [18]. As Serra [18] points out, weather, food, and wine are some of the most noticeable traits of the Algarve region and are a source of tourist interest (2021).

The article is organised as follows: (1) the next section will summarize the European Union's efforts to promote sustainable tourism since 1993; (2) the subsequent section ex-

plains the TARGET_TIA methodology used to measure the impacts of POSEUR (2014–2020) in this article; (3) next, the discussion section dissects the issues related to the implementation of POSEUR and analyses the benefits that the operational programme brought to the Algarve region; and finally, (4) the conclusion provides insights produced by our study that can be helpful for future operational programmes in the domain of sustainable tourism.

## 2. Literature Review/Theoretical Background

The development of the concept of sustainable tourism gave rise to an international legislation framework, of which the first milestone was the publication of the Charter for Sustainable Tourism, originally drawn up in 1995 [19]. This document resulted in "a broad framework for local-scale sustainable development of tourism by listing several objectives related to the social, economic and environmental sustainability of the phenomenon." [20]. In the subsequent years, the United Nations (UN) published the Global Code of Ethics for Tourism [21]. In 2004, the World Trade Organisation (WTO) officially designated the variables that are important to analyse when looking at sustainability in the tourism sector by releasing the *Guidebook—Indicators of Sustainable Development for Tourism Destinations* [22]. Some of those indicators are: (i) the environmental impacts of tourism activity; (ii): the dependence on natural resources; (iii) responses from tourism companies to environmental challenges; and (iv) the socioeconomic implications or the importance of the infrastructural conditions [23]. The EU legislation followed the international frameworks and, in 2001, they released the communication: "A cooperative approach to the future of tourism" [24]. Agenda 21 for a Sustainable and Competitive European Tourism [25] and the new framework for European Tourism [26] were pivotal to the introduction of sustainability awareness in EU legislation. Therefore, growth in the sector brought new paradigms to this market, such as the risks in coastal areas, which were regulated by the European strategy for coastal and maritime tourism [27].

To invest in sustainability in tourist areas, it is important to comprehend the underlying structure that impacts the functioning of tourism over time [28]. There is also a need to find a balance between the economic, sociocultural, and environmental characteristics of the territory and the development of the concept of sustainability [5,29–31]. Portugal has also been developing several plans for environmental management in recent years to promote a sector of activity that will be capable of achieving its full potential regarding the natural and cultural assets of the country [32].

The Action Plan to Assist Tourism 1993–1995 [33], the Charter for Sustainable Tourism [19], the Green Book named *The Role of the Union on Tourism* [34], and the Philoxenia Programme [35] initiated the institutional development of tourist activity inside the EU [36]. The Action Plan to Assist Tourism [33] and the COUNCIL DECISION 92/421EEC approached the matters of the transport sector and the pivotal role of local decision making for sustainability for the first time. As previously mentioned, the Charter for Sustainable Tourism created awareness of the importance of balancing tourist activity with measures that can reinforce environmental protection [20]. The Green Book, *The Role of the Union on Tourism* [34], discussed the institutionalization of tourism and launched the foundations for the Philoxenia Programme [35] to stimulate knowledge in the tourism sector.

EU legislation made use of the above guidelines with the release of the communication "Working together for the future of European Tourism" [24], which was followed by the Council Resolution on 21 May 2002 [36], which aimed at strengthening small- and medium-sized tourism enterprises in Europe. This was followed by the establishment of Agenda 21 for a Sustainable and Competitive European Tourism [25], 21 points that outlined the need for the efficient development of the tourism sector. Almost simultaneously, three conferences were held by the EU to discuss different topics inside the tourism domain: on 8 September 2005 about new perspectives and challenges to European tourism [37]; on 29 November 2007 to approach "A new policy to a renewal of European tourism: towards a reinforced partnership to European tourism" [38]; and on 16 September 2008,

the LITTORAL conference was held to debate the impact of tourism in the development of coastal areas [39].

The EU Cohesion Policy phase, initiated in 2007, was fundamental in the development of sustainable tourism at the local level, and European Regional Development Funding allocated an important part of the EU's budget to the Cohesion Policy [40], which "the Member States implemented . . . through regional projects" [41]. The Framework for European Tourism of 2010 [26] contributed to this process of decentralizing tourism activities and included new strategies that gave it a more institutional framework [41]. Examples of those strategies are the "Maritime Integrated Strategy", the "Multiannual Financial Framework", the "Virtual Tourism Observatory", and the "European Job Mobility Portal" (idem). Soon after, the commission launched ETIS, a system to measure the effectiveness of tourist destinations and which evaluates regions based on four categories: "(a) Destination Management, (b) Social and Cultural Impact, (c) Economic Value and (d) Environmental Impact" [42].

The POSEUR programme was one of the initiatives financed via the framework of the European Cohesion Policy, and directly impacted the progression of sustainable tourism in Portugal. More recently, a cooperation agreement was agreed between the UNWTO and the European Union to establish a partnership for the promotion of sustainable tourism (2018). In 2020, the European Parliament approved the resolution on "tourism and public transportation in 2020 and beyond" [43], which stands for the modernization of the public transport sector to become more sustainable and accessible to people with disabilities. In addition, some rewarding initiatives such as Eden [44], COSME [45], the Green Belt [46], and the EuroVelo [47] helped tourist enterprises to implement more sustainable practices in different domains. Eden and the COSME are focused on the promotion of European destinations that accomplish proficient sustainability ratings, the Green Belt network contributes to the protection of natural areas, and the EuroVelo programme incentivizes the development of bike lanes and bike usage in the EU.

Considering the sustainability performance in the tourism sector inside the EU, Lozano-Ramirez et al. [23] point out countries such as Spain, Malta, and the Netherlands as top performers inside the European framework. On the other hand, Greece and Finland were considered the least attractive for tourists in terms of environmental protections. However, the tendency is for EU destinations to start to be more aware of the harmful effects of tourist practices on the environment, and sustainability indicators are changing at the local level due to the cohesion policies [41]. More crucially, the EU Cohesion Policy has brought benefits to EU sustainable tourism at national, regional, and local levels since there is a general sense of satisfaction with the tourism infrastructures; therefore, the image of several EU countries was amplified and the traffic related with tourism activity has been managed with more efficient tools [48].

Therefore, institutional efforts have been made to assure that there are positive developments in sustainable tourism in the EU. According to the literature, the goal is to tackle issues that are highly relevant to the literature in this field of study, as suggested by [49] (2022), such as the adoption of renewable energies [50,51], mitigating greenhouse gas emissions and the pollution caused by the transportation industry [52,53], improving waste management [54,55], and alleviating water scarcity [56,57] and the risks and vulnerabilities caused by climate change [58,59]. Some of the recent solutions for the challenges mentioned above include the construction of tourism infrastructures with renewable materials [60], the promotion of cycling as a mobility method in touristic urban areas [61], and the reuse of water in resorts [57], as well as the adoption of seawater desalination to irrigate public areas [62]. Most of these sustainability measures were implemented in European countries due to the financial conditions made available by the EU Cohesion Policy. In the Algarve region, changes occurred between 2014 and 2020 through the POSEUR programme, which allocated EU funds to regional stakeholders. The most easily identifiable changes towards a more sustainable model of tourism in the Algarve were related to new methods of water management on golf courses [63], the renewed protection of coastal areas [1], and invest-

ment in environmentally friendly and cleaner solutions in the transport sector, all of which were based on EU sustainability principles.

## 3. Materials and Methods

Some studies focused on the need to find indicators that can properly evaluate the development of tourism sustainability in specific regions [42]. Simultaneously, the importance of incorporating different stakeholders (public entities at international, national, and local levels, as well as private enterprises) to define those indicators brings new challenges to the measurement of the impact of environmental policy [64,65]. In this study, we applied a territorial impact assessment (TIA) methodology called TARGET_TIA [66] to assess the main impacts of POSEUR in the Algarve. The methodology we used is different from the ESPON_TIA model due to its higher flexibility and reliability in evaluating the impacts in the various stages of the respective policy/project/programme [6]. In this study, we conducted the analysis to understand the ex post impacts of POSEUR on the tourism activity in the NUTS II Algarve. The data inputted in the TARGET_TIA model were based on statistical environmental indicators of public access and five interviews with stakeholders of the region: (1) Águas do Algarve, (2) ALGAR, (3) Comissão de Coordenação e Desenvolvimento Regional do Algarve, (4) Câmara Municipal de Faro, and (5) Prof. Thomas Panagopoulos from the University of Algarve. This model of measurement makes use of concepts such as regional sensibility, policy intensity, and causality (Figure 1).

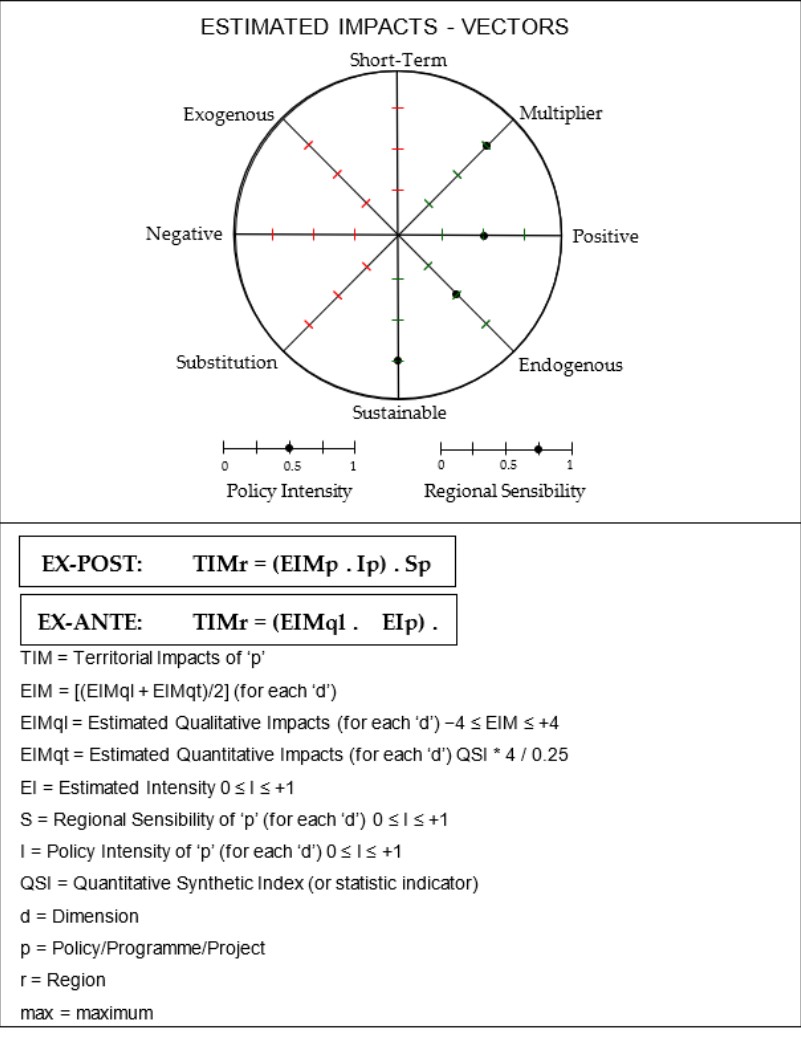

**Figure 1.** TARGET_TIA ex ante and ex post formulas. Source [66].

In essence, the TARGET_TIA produces impact scores for each dimension of analysis resulting from the collection of qualitative (interviews + literature review) and quantitative (statistics + project analysis) data. The interviews are crucial since quantitative data is insufficient to obtain an accurate picture of the potential impacts of any project, programme, or policy. Only deeply involved stakeholders have a detailed perspective on the causality and impacts of public investments. Here, rather than conducting vast quantities of interviews, we felt the best strategy was to contact a few interviewees with in-depth knowledge of the impacts of the evaluated subject. Alongside their opinion, an extensive literature review is crucial to assess the regional sensibility towards the evaluated investments. Following the TARGET_TIA methodology, a minimum score of 0 was given when the region does not require more investments in the analysed policy for its positive territorial development trends. Conversely, a maximum score of 1 is attributed when such investments are imperative for its development trends. In addition, we used quantitative data related to the project databases (investment per analytical dimension) to assess the policy intensity element of TARGET_TIA. Here, a maximum score of 1 is given when an analytical dimension is highly financed, which would maximize its potential impacts on the analysed territory. The opposite is true when there is limited investment dedicated to a specific analytical dimension. Finally, we used statistical data to analyse the development trends of the territory under study during the period of the project–programme–policy evaluation. The main goal is to detect potential causalities of the analysed subject in each analytical dimension.

For the case of POSEUR in the Algarve, we collected qualitative data via five interviews (one academic and four stakeholders), and we defined the regional sensibility scores mostly by reading the most recent (2015) national spatial planning reports. Firstly, we obtained the quantitative data by analysing the Portugal 2020 project database. Here, we divided the projects into the five selected analytical dimensions to see the respective percentage of the total funding of POSEUR (2014–2020) in the Algarve. We adjusted the respective policy intensity scores by these percentages. Finally, we collected the statistical data via national statistics. Specifically, we used the following indicators (one for each of the five analysed dimensions): (i) electricity production from renewable energy sources through new technologies (total) MW; (i) municipalities' environmental expenditures per capita (protection of air quality and climate) (€); burnt area %—rural fires lasting more than 24 h (no.); municipalities' environmental expenses per capita (protection of biodiversity and landscape); where more and less garbage is selectively collected, on average, per person—kg/inhabitant ratio.

*3.1. Regional Sensibility*

The regional sensibility levels are influenced by the demographic, political, infrastructural, and economic conditions of the Algarve region [67–69]. If the preconditions of the region lack efficiency or present environmental risks to the future of the region, the regional sensibility value will be closer to 1. The more effective the preconditions, the closer the value will be to 0. We defined the sensibility for the region of the Algarve by the analysis of documents that approached the environmental issues of tourism in the region, such as the PROT Algarve—the first report on the Regional Spatial Planning Plan [70]—and PNPOT [8].

The Algarve region has suffered from low rain precipitation years, which led to severe dry periods, contributing to the degradation of agricultural land [70]. Under these circumstances, unsustainable tourism activity is seen as an additional problem that undermines the quality of the soils in the Algarve [71]. This phenomenon has a direct influence on the shortage of water resources, which is not only troubling to the economic activity in the Algarve but also creates difficulties and new challenges for the fire prevention mechanisms in the region [72]. Additionally, there is a new source of regional sensibility resulting from the overcrowding in the Algarve's coastal areas, which causes the degradation of dunes from coastline urbanization and the erosion of green spaces [1]. Flood risks are also a

reality in the Algarve region; the streams of water can easily reach villages due to the characteristics of the riverside, mostly constituted by sandy barriers [73].

Moreover, there is an excessive usage of private vehicles in the most densely populated areas of the Algarve; this is part of the cause and an effect of the problems in the public transportation system and its infrastructures [74]. The growing geriatric population visiting Algarve produces the need to promote sustainability and more efficient public transportation methods that can serve the more socioeconomically fragile population [74,75]. The rise of some issues related to the inefficiency of the natural protected areas in the Algarve is also a reason for concern. Pallero et al. pointed out that the "( . . . ) non-implementation of the two initiatives to protect Guadiana under a Natural Park" [76] is crucial for comprehending the lack of adequacy of public policy regarding ecosystem dynamics. The same can be said about the quick expansion of golf courses, with some of those being licensed to operate in areas protected by the *Natura 2000* network, which confirms the tendency to build these leisure sites in unsuitable locations for tourism activity [13].

### 3.2. Policy Intensity

The TARGET_TIA methodology applied in this research used a policy evaluation element called policy intensity, which supports an assessment of the causality between the effective impacts of public policy and financial resources used in each analysed parameter [77]. In this variable, a high amount of funding made available by the political institutions for sustainability in the Algarve region will correspond to a higher value of political intensity. In the Algarve, there was a total POSEUR investment of €72,282,693.00 (Table 1):

**Table 1.** POSEUR investment in the Algarve by analytical dimensions.

| Dimensions | Algarve |
|---|---|
| A: Low-Emissions Economy | €1,902,925.00 |
| B: Climate Change Adaptation | €5,837,162.00 |
| C: Risk Prevention and Management | €7,145,161.00 |
| D: Environment Protection | €9,283,633.00 |
| E: Resource Efficiency | €48,113,812.00 |
| Total | €72,282,693.00 |

Source: POSEUR projects—funding allocated by dimension in the Algarve.

### 3.3. Causality and Impact Scores

The funding made available by POSEUR concerning the dimensions that directly impact tourism in the Algarve were dedicated to: (1) the renewal of the public transportation systems with electric vehicles, mainly in the regions of Portimão and Albufeira; (2) the restructuring of damaged areas to accomplish biophysical stability and a balanced ecosystem, with special emphasis on the project in the Sagres Fortress; (3) reinforcing the most fragile coastal areas to prevent risks in the areas of Tavira, Monte Gordo, and Lagos; (4) improving the general quality of the beaches with an investment mainly directed towards providing infrastructures, such as walkways on the coastline; (5) the protection of valuable natural resources by reinforcing the natural protected areas (e.g., Olhão) and propagating information about the fragility of the territory; and (6) the construction of new infrastructures dedicated to water treatment, such as Companheira and Faro-Olhão ETAR. Therefore, this study aims to understand if this investment established a causality relation with a real impact on the tourism industry of the Algarve region. The impact scores presented in Table 2 reflect the causality relation on a scale between −4 (impact significatively prejudicial) and 4 (impact significatively positive).

**Table 2.** POSEUR evaluation impact matrix—NUTS II Algarve.

| Dimensions | Impact Scores (−4/+4)/Contrafactual | | | | | Tuning Elements (0–1) | | Causality—Territory Features (0–1) | | Impact (score) |
|---|---|---|---|---|---|---|---|---|---|---|
| | Pos/Neg | End/Exo | Sus/Cur | Mul/Sub | Mean | Int/Pol | Sen/Reg | 2014 | 2020 | (−4/+4) |
| Low-Emissions Economy | 3 | 3 | 3 | 3 | 3 | 0.5 | 0.5 | 0.25 | 0.25 | 0.375 |
| Climate Change Adaptation | 3 | 3 | 3 | 3 | 3 | 0.75 | 0.75 | 0 | 0.5 | 3.094 |
| Risk Prevention and Management | 3 | 3 | 3 | 3 | 3 | 0.5 | 0.75 | 0.25 | 0.25 | 0.563 |
| Environment Protection | 3 | 3 | 3 | 3 | 3 | 0.25 | 0.5 | 0.25 | 0.75 | 0.688 |
| Resource Efficiency | 3 | 3 | 3 | 3 | 3 | 0,5 | 0,75 | 0.5 | 0.75 | 1.313 |
| Mean | 3 | 3 | 3 | 3 | 3 | 0.5 | 0.5 | 0.25 | 0.25 | 1.138 |

Note: Pos/Neg: positive vs. negative; End/Exo: endogenous vs. exogenous; Sus/Sho: sustainable vs. short-term; Mul/Sub: multiplier vs. substitution; Pol/Int: policy intensity; Reg/Sen: regional sensibility. Source: own elaboration.

## 4. Discussion and Results

Stakeholders claim that POSEUR has facilitated the quick implementation of some projects, even if there were projects that did not reach their desired goals. One of the fragilities of the programme was leaving the theoretical plans and moving towards more concrete action. For the stakeholders, the goals were reached in overall terms. However, some stakeholders consider that the goals were not ambitious enough. Most stakeholders also pointed out that the local actors are not as dynamic as in other regions, which is very important as it is crucial for the stakeholders to have a clear "shared vision of the issues and priorities on sustainable tourism development to assess the competitiveness of the region" [5]. In this regard, Wanner et al. [78] argue that stakeholders' opinions are fundamental to the planning process.

The Algarve region has been financially penalised as it belongs to the transition group of EU regions, largely due to the wealth created by tourist activity. This results in limited funding to support public policies when compared with other regions in mainland Portugal. Hence, the stakeholders showed concern regarding the amount of funding available to the Algarve region. They considered it insufficient when compared with other Portuguese regions, mainly because the inhabitants' wealth is not comparable to the wealth of the tourists that boost the local economic indicators. However, this tourism activity has a seasonal dynamic, which leaves the inhabitants of the region suffering during the low season from the inflated indicators reached in the high season. A representative of the municipality of Faro mentioned that it would be easy to implement EU-financed projects if there were sub-programmes that could help their financial execution. In all, POSEUR was aligned with the guidelines of the national public policy. However, it was not enough to promote a systemic change in the region when it came to its contribution to more sustainable tourism. For instance, there were problems in the preventive actions of POSEUR to ensure that "over-tourism was not affecting the local ecology" [79].

### 4.1. Tourism and the Low-Emissions Economy

The interviewed stakeholders gave special relevance to the impacts of POSEUR regarding the Plan for Adaptation to Climate Change, mostly because it has supported the creation of more green public spaces. This dimension was the least impacted by POSEUR in the Algarve region, with an impact score of 0.375. There is still a long path to travel concerning the low-emissions economy and the mitigation of risks in coastal areas; for example, golf courses are still not achieving the desired numbers of $CO_2$ emissions [80]. Besides that, there is still potential in the region to achieve cleaner energy [81]. The ALGAR association kept the typology of the waste collection vehicles but with renewed engines, so with less emissions, even though diesel is still the main fuel. The replacement of the engines for an *euro6* model allows a less harmful practice concerning emissions levels and has better environmental performance. However, the ALGAR association mentioned that this did not have an influence on municipalities' economic fuel consumption.

Crucially, the stakeholders consider the impact of POSEUR as medium positive in this analytic dimension. Moreover, there is also a promise that there will be a reinforcement of new solutions for cleaner mobility in the next operational programmes mostly in Faro, Loulé and Olhão, highly populated cities that have an active tourist sector. The municipality of Faro sees an improvement of the energetic efficiency and an increment of renewable sources of energy in the Algarve due to actions within POSEUR. These directly impact the awareness to de-carbonize human activities in the region. Sensibilization campaigns for greater uptake of electric mobility in the public opinion were also widely spread during this period.

### 4.2. Tourism and the Adaptation to Climate Change

The entities interviewed defend that the difference between hydric availability and resources before and after the POSEUR programme is noticeable. There is a new sense of regional and local awareness of climate change, as well as an attempt to preserve the morphologic characteristics of the region. The stakeholders gave special relevance to the progress guaranteed in the climate change adaptation dimension, as the development of climate change at a frenetic pace in the area [82] dramatically changed the content of the projects that were presented after 2014, which made it harder to exploit the landscape without damaging the environment [83].

It must be emphasized that cities in the Algarve have a problem of resilience concerning climate change, as there is a lack of long-term integrated strategies for the development of plans that can mitigate the environmental damage caused by climate change [84]. Therefore, adaptation to climate change was the dimension with the highest impact score, mostly due to the importance of the hydric availability issue and the lack of previous plans to mitigate the effects of climate change in the territory. Relevance was also given to the PAACs (Plans for Adaptation to Climate Change) which produced the guidelines on the vulnerabilities, projections and measures for the next years in the Faro municipality and other cities in the Algarve. In sum, POSEUR action for climate change adaptation contributed to the possibility of creating research studies and supporting plans for future decision-making in the tourism sector.

### 4.3. Tourism and Risk Prevention and Management

There should be a more strategic vision about the deficiencies of the territory; for example, the desertification process lacked attention [85]. POSEUR lacked initiatives concerning the rising sea level in the Algarve [15], which poses a significant risk to the regional economy since the bulk of the region's economic income and demography is located near the coast. The preventive action was not sufficient, and this is not the first programme in which the Algarve region is looking at the problems only when they have already happened, as the interviewed entities noticed. This can explain the low impact score obtained by the implementation of POSEUR in the Algarve in this dimension, since some of the structural problems of the region are still to be handled in terms of risk prevention and management [86,87]. In this context, the interviewed stakeholders concluded that it is important to invest in the reduction of the amount of waste that ends in landfill sites, as well as to close some of those landfills. A considerable number of water treatment stations still discharge their content directly into the environment, which pollutes the region's rivers and water resources. Alongside this, it was mentioned in the interviews that deactivated mining sites still threatened the environment by exposing the region to uranium levels. Contrary to what would be expected, POSEUR did not contribute to solving this problem, even though it poses a risk to the population.

On the positive side, the constitution of the Plan for Adaptation to Climate Change (PAAC) in Faro allowed the production of new emergency plans, information, and sensibilization campaigns, as educating the population is fundamental for a sustainable transition [88], as well as action towards the empowerment of local actors. On the other hand, the members of the municipality of Faro, when interviewed, pointed out the relevance

of providing prevention of catastrophic events in the Faro area, even if there are some tools missing for more effective action. Finally, it is important to mention that the ongoing expansion of green spaces and shadow areas in Faro's downtown area is important to tourist activity. Some of these were supported by POSEUR investment.

*4.4. Tourism and Environmental Protection*

In this dimension, the impact score reveals that the results were short of initial expectations. However, the interviewed entities reinforced the idea that POSEUR helped to bring to light plans of environmental protection that were already implemented but not working to their full potential. Indeed, almost 70% of the Algarve territory is under some sort of environmental protection status, including the *Natura 2000* network. As such, POSEUR helped to keep these areas under protection. Alongside this, the delimitation of the management area is fundamental for the tourism sector as it contributes to "promoting and supporting sustainable development" [76]. The CCDR Algarve points out that the record number of beaches classified with blue flags in the Algarve region would not be possible without the effort of POSEUR in the sector of water treatment. The coastal area underwent several interventions to stabilize the sands and protect the tourist sector and other activities inherent to it by incrementing the level of protection on beaches in the Algarve territory. The growing awareness of the population regarding the efficient use of water was also mentioned by some stakeholders. More information means more sensitivity in terms of environmental risk [11].

The interviewed entities support the idea that this awareness of the decision-making actors regarding the need to improve environmental quality helped to bring the vulnerabilities of the territory under the protective tools of territorial management. The phenomenon most impacted by these new management tools is the excessive consumption of resources and the ineffective mitigation of waste production. In this sense, the diverse ongoing sustainability programmes in the Algarve leveraged green areas in public spaces and were a fundamental help in implementing changes in the public transportation sector. Furthermore, the option to strengthen the public transport sector with greener vehicles as well as the raised awareness of the population regarding cleaner means of transportation and the proliferation of new bike lanes have the potential to positively impact the tourism sector and inhabitants' quality of life in the Algarve.

*4.5. Tourism and Resource Efficiency*

The resource efficiency dimension had two strategic areas of intervention in the Algarve. Firstly, the construction of two ETARs, as specific sites dedicated to water treatment—one in Companheira (near Portimão) and the other in Faro-Olhão—helped the region to reach numbers close to 100% in water treatment quality. This infrastructural improvement helped to solve systemic situations in which the region had already been penalized for the environmental harm caused by a lack of appropriate green infrastructure. Needless to say, this investment has a direct causality on the general quality of Algarve beaches, in particular in the Ria Formosa (close to Faro). As Lukoseviciute and Panagopoulos mentioned: "Beach management aims to satisfy users, but it should also address education and raising awareness of environmental values and global climate change issues" [1]. The construction of the ETARs was an emblematic milestone for the sanitation cycle, as pointed out by the CCDR Algarve. Other projects dedicated to water reuse, such as connections between larger ETARs and smaller stations of water treatment, were also funded by POSEUR.

The Algarve region saw a significant loss in tourism activity during the COVID-19 pandemic years [31]. However, those were years of intense climatic dryness that forced the Algarve NUTS II to use its reservoirs to their maximum capacity. Therefore, concern over the processes of water scarcity resulted in a pilot project to be implemented during the following POSEUR in Vila Real de Santo António, near the Spanish border. This project aims at promoting alternative uses of the water, before returning it to the open-air environment. The main goals of the projects in the Algarve related to water treatment are a

reduction in water loss, a renewed awareness of the efficient use of water, irrigating larger green areas with less water, and maximizing the efficiency of golf course watering [13]. On the other hand, there was a strong investment in the waste management system, which was quickly modernized by POSEUR. Waste management processes are fundamental in establishing a circular economy in the Algarve [89] and during the period of POSEUR, the region reached numbers close to 40/50 hundred thousand tons of recycled waste processed each year. During their interview, the representative of the Commission for the Regional Development of Algarve mentioned that the amount of recycled waste in the Algarve is three percentage points above the rest of the country. Portugal is a success case because it moved from a traditional system to a modernized cycle of waste management, as pointed out by the ALGAR association. Although, as identified by the interviewed entities in prior dimensions, what has been performed to reduce the amount of waste that ends in landfills is not sufficient. There is a need to do more to reduce that waste and to reduce landfill sites.

The impact scores obtained for the resource efficiency dimension consider the total alignment of POSEUR with the goals of sustainable development in the Algarve. According to the interviewed stakeholders, there is no guarantee that the projects will later be executed as described in the projects. In addition, there are peculiar traits in the Algarve region that can be regarded as an obstacle to the efficiency of its resources, such as the infrastructures that are built to be used at the peak of tourism activity, which means that those same structures will suffer from underuse in the low seasons.

The low positive scores obtained in four of the five dimensions of the analysis do not have a causal relation with the amount of investment made available by POSEUR, as the dimension with the highest impact score (climate change adaptation) had the second lowest investment of all dimensions allocated to it. Simultaneously, the resource efficiency dimension, which obtained the highest investment in the Algarve region, did not reach the expected outcome, with an impact rate of 1.313. The different needs of the region can explain this heterogeneity of results, as most of the tourism sustainability issues were approached at a superficial level, and the same structural problems continued to persist even if mitigated by the European Cohesion Policy. The high degree of sensibility in the region partially explains the slow pace of change towards a more sustainable model of tourism. However, the interviewed entities were in agreement that the next operational programme 2021–2027 and the regional programmes associated with it have the potential to boost the foundations launched by POSEUR (2014–2020).

Overall, the implementation of POSEUR fell short when compared with our initial expectations; nonetheless, it was aligned with the European Cohesion Policy guidelines for sustainable tourism by promoting cleaner means of mobility, supporting R&D initiatives that resulted in the elaboration of different Plans for Adaptation to Climate Change, creating more green public spaces, and keeping the beaches' quality while preventing coastal erosion. Nonetheless, the small extent of these changes is reflected in the low impact scores, which alerts the European Union and the European Cohesion Policy to the need to focus on the challenges posed by the different regions in European countries. It is even more important to look at the characteristics of the regions when they are highly impacted by tourism, which increases the sensitivity of these regions to the economic and environmental impacts of this sector of activity.

## 5. Conclusions

The main goal of this paper is to assess the main impacts of POSEUR (2014–2020) on sustainable tourism in the Portuguese NUTS II region of the Algarve, looking at five different analytic dimensions: (1) a low-emissions economy, (2) adaptation to climate change, (3) risk prevention and management, (4) environmental protection, and (5) resource efficiency. A TIA methodology was used (TARGET_TIA), utilizing a mix of qualitative (interviews + literature review) and quantitative data (project and statistical databases). With the application of this methodology, it was concluded that, in overall terms, POSEUR had a reduced positive impact on the region for improving environmental sustainability

processes. Crucially, the amount of funding available was insufficient to promote a systemic change in the region towards a sustainable model of tourism. There are, however, some positive impacts resulting from the implementation of POSEUR in such areas, namely the improvement of water efficiency in golf courses, the renewed coastal areas adapted to the risks of the ecosystem, and the promotion of cleaner methods of public transport. Nevertheless, none of those initiatives reached the impacts obtained in the dimension of adaptation to climate change. The main reason was the support of POSEUR given to launch the PAAC plans that set the guidelines for future sustainability decision making in various environmental areas of the Algarve. Furthermore, some environmental sectors in the region need further attention from the European Cohesion Policy, such as for the reduction of emissions in tourist activities, the mitigation of the effects caused by landfills and their waste, and attention to the deactivated mining sites, which are still damaging the region's environment, as well as the need to balance the infrastructure efficiency between the high and low seasons.

It goes without saying that the research found some limitations, including the limited number (five) of interviewees and the respective content, which is highly dependent on their opinion and vision. Despite these limitations, the collected information is robust enough to support the conclusions since key regional stakeholders and a top academic were interviewed. In the future, it would be interesting to follow this line of research and compare the impacts of European Cohesion Policy-financed projects on sustainable tourism in several EU touristic areas, as well as its impacts on inhabitants' quality of life.

Ultimately, the realisation of these impact assessment analyses can provide crucial insights to improve EU, national, and regional sustainable tourism strategies towards increasing the efficiency and effectiveness of public funding. Crucially, the funding made available as well as the nature of the programmes must be adapted to the environmental, economic, social, and structural characteristics of the regions. Here, the empowerment of regional stakeholders is pivotal for the sustainable development of the local economy.

**Author Contributions:** Conceptualization, E.M.; methodology, E.M., B.V.; formal analysis, E.M. and B.V.; investigation, B.V.; resources, E.M. and B.V.; data curation, B.V.; writing—original draft preparation, E.M. and B.V.; writing—review and editing, B.V. and E.M.; visualization, B.V.; supervision, E.M.; project administration, E.M.; funding acquisition, E.M. All authors have read and agreed to the published version of the manuscript.

**Funding:** POAT-01-6177-FEDER000063-Monitoring and assessment of territorial impacts of public policies via a WEB-GIS platform.

**Institutional Review Board Statement:** Not applicable.

**Informed Consent Statement:** Not applicable.

**Data Availability Statement:** Not applicable.

**Conflicts of Interest:** No conflict of interest.

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
