# Peer review of "The Impacts of EU Cohesion Policy on Sustainable Tourism: The Case of POSEUR in Algarve"

_sustainability, doi:10.3390/su141912672_

Round 1
Reviewer 1 Report
As written in conclusion, this paper has assessed the impacts of sustainable tourism in the Portuguese with five different analytic dimensions: such as 1) economy with low emissions, 2) adaptation to climate change, 3) risk prevention and management, 4) environmental protection and 5) resource efficiency.
I can judge the research methods are well established and results and discussion are also appropriate.
Author Response
As written in conclusion, this paper has assessed the impacts of sustainable tourism in the Portuguese with five different analytic dimensions: such as 1) economy with low emissions, 2) adaptation to climate change, 3) risk prevention and management, 4) environmental protection and 5) resource efficiency.
I can judge the research methods are well established and results and discussion are also appropriate.
R: Many thanks for the positive remarks
Reviewer 2 Report
The reviewer agrees with the authors that understanding the impact of EU cohesion policy on sustainable tourism is critical for the future development of a sustainable tourism industry. This is a timely and meaningful study and its finding could help the industry better understand the current situation and lay a foundation for future research. However, the reviewer believes there is room for improvement.
1) The Introduction section (section 1) and EU Funding and Sustainable Tourism section (section 2) could be restructured into two different sections: Introduction and Review of Literature. The background information about EU cohesion policy, EU funding, and tourism in the Algarve NUT2 region could all be included in the Introduction section. The authors have provided very detailed and thorough background information. However, the reviewer believes the background information could be a little concise because not all information in the manuscript seems closely and directly related to the study. The Review of Literature section could include sustainable tourism related literature (e.g. what has been done, what are the trends, what issues have been identified, and etc.). It is particularly important to more thoroughly explain and discuss the selection of the five-dimension measuring model as it is the core of this study.
2) The Methodology section needs to be enhanced. Specifically: i) The authors might want to create different subsections to more clearly explain the methods (e.g. study design, sampling, data collection, data coding, data analyses, and etc.). ii) The TARGET_TIA model needs to be more thoroughly explained to justify why it fit this study (i.e. Why other models such as ESPON_TIA did not fit?), iii) Data need to be very clearly described. Particularly, since the authors clearly mentioned that both qualitative and quantitative data were used, it is important to clearly explain, define, and distinguish. iv) The development of the survey instrument needs to be more clearly explained. v) Exactly how did the five interviews answer the research questions? Were there any themes identified from the interview?
3) The TARGET_TIA is not a research method. It is a model/theory developed in another research project. Exactly what research method was used in this study? Qualitative, quantitative, or mixed-methods?
4) The reviewer believes that, to make the study more meaningful and more relevant to the industry, a more detailed outlook of the tourism industry in Algarve should be discussed with focus on sustainability and EU cohesion policy based on the findings of this study.
Last but not the least, the reviewer would like to suggest a thorough English proofreading and editing. Many sentences need to be re-written or re-structured to improve the overall readability of the manuscript.
Author Response
Rev2
The reviewer agrees with the authors that understanding the impact of EU cohesion policy on sustainable tourism is critical for the future development of a sustainable tourism industry. This is a timely and meaningful study and its finding could help the industry better understand the current situation and lay a foundation for future research. However, the reviewer believes there is room for improvement.
A - The Introduction section (section 1) and EU Funding and Sustainable Tourism section (section 2) could be restructured into two different sections: Introduction and Review of Literature. The background information about EU cohesion policy, EU funding, and tourism in the Algarve NUT2 region could all be included in the Introduction section. The authors have provided very detailed and thorough background information. However, the reviewer believes the background information could be a little concise because not all information in the manuscript seems closely and directly related to the study. The Review of Literature section could include sustainable tourism related literature (e.g. what has been done, what are the trends, what issues have been identified, and etc.). It is particularly important to more thoroughly explain and discuss the selection of the five-dimension measuring model as it is the core of this study.
R: These issues were addressed in the revised version
B - The Methodology section needs to be enhanced. Specifically: i) The authors might want to create different subsections to more clearly explain the methods (e.g. study design, sampling, data collection, data coding, data analyses, and etc.). ii) The TARGET_TIA model needs to be more thoroughly explained to justify why it fit this study (i.e. Why other models such as ESPON_TIA did not fit?), iii) Data need to be very clearly described. Particularly, since the authors clearly mentioned that both qualitative and quantitative data were used, it is important to clearly explain, define, and distinguish. iv) The development of the survey instrument needs to be more clearly explained. v) Exactly how did the five interviews answer the research questions? Were there any themes identified from the interview?
R: These issues were addressed in the revised version
C - The TARGET_TIA is not a research method. It is a model/theory developed in another research project. Exactly what research method was used in this study? Qualitative, quantitative, or mixed-methods?
R: This issue was addressed in the revised version
D - The reviewer believes that, to make the study more meaningful and more relevant to the industry, a more detailed outlook of the tourism industry in Algarve should be discussed with focus on sustainability and EU cohesion policy based on the findings of this study.
R: This issue was addressed in the revised version
E - Last but not the least, the reviewer would like to suggest a thorough English proofreading and editing. Many sentences need to be re-written or re-structured to improve the overall readability of the manuscript.
R: This issue was addressed in the revised version
Reviewer 3 Report
Dear Authors,
The paper entitled "The impacts of EU cohesion policy in sustainable tourism: the case of the PO SEUR in Algarve" aims to better understand the impact of public investments financed by EU cohesion policy in the tourism sector in the Portuguese Algarve.
The research topic is very important due to the ongoing climate change also influenced by tourism.
The work presented for evaluation is a review, with a small empirical part.
After reading the work, I have the following comments and suggestions for improving the work:
Titel
The title of the work should be modified. I suggest adding at the end of the title in "Portuguese Algarve".
Structure of the article
I suggest improving the structure of the article according to the guidelines of the Journal "Sustainability"
New numbering of chapters should be introduced:
1. Introduction
2. Literature review/theoretical background
3. Materials and methods
4. Results
5. Discussion
6. Conclusion
In the Introduction
The introduction to the topic is interesting and based on the world literature. At the end of this chapter, the purpose of the work was placed, but the research questions were missing.
Chapter 2 EU funds and sustainable tourism
I suggest renaming it to: Literature review/theoretical background
Chapter 3 Methodology/ Impact of POSEUR in the Algarve
I suggest changing to: Materials and methods
I feel that some of the information contained in this chapter should be moved to the chapter Results a Which is missing from the article.
Chapter Results
I suggest presenting the results of the surveys
In the Discussion section, the authors should discuss and explain the findings and results of the paper more. This would contribute to the high improvement of the paper. The authors should compare their project and results with the results of similar studies on this topic from other countries in the world.
Technical errors
Incorrect notation of literature - literature should be written in the text as a number, e.g. (14)
[102-104]- unintelligible abbreviations (I), (II)
Figure 1 TARGET_TIA ex-ante and ex-post formulas.-I suggest improving the quality of the figures
In conclusion, I strongly recommend this paper for publication in the journal Sustainability after making significant changes.
Regards
Reviewer
Author Response
Review 3
A - The title of the work should be modified. I suggest adding at the end of the title in "Portuguese Algarve".
R: There is only one Algarve. Adding Portugal is a redundancy.
Structure of the article
B - I suggest improving the structure of the article according to the guidelines of the Journal "Sustainability"
New numbering of chapters should be introduced:
- Introduction
- Literature review/theoretical background
- Materials and methods
- Results
- Discussion
- Conclusion
C – Introduction: The introduction to the topic is interesting and based on the world literature. At the end of this chapter, the purpose of the work was placed, but the research questions were missing.
Chapter 2 EU funds and sustainable tourism
I suggest renaming it to: Literature review/theoretical background
Chapter 3 Methodology/ Impact of POSEUR in the Algarve
I suggest changing to: Materials and methods
I feel that some of the information contained in this chapter should be moved to the chapter Results a Which is missing from the article.
Chapter Results
I suggest presenting the results of the surveys
R: Issues addressed in the revised version
Technical errors
Incorrect notation of literature - literature should be written in the text as a number, e.g. (14)
R: If the paper is accepted for publication that will be done.
Figure 1 TARGET_TIA ex-ante and ex-post formulas.-I suggest improving the quality of the figures
R: Done.
In conclusion, I strongly recommend this paper for publication in the journal Sustainability after making significant changes.
- many thanks for the useful comments
Round 2
Reviewer 3 Report
Dear Authors,
YOUR PAPER HAS BEEN THOROUGHLY REVISED IN ACCORDANCE WITH THE RREVIEWERS SUGGESTIONS AND DIRECTIONS.I MAKE NO FURTHER COMMENTS .I RECOMMENT THE PAPERTO THE SUBMISSION.